# A Pseudo-labeling Approach to Semi-supervised Organ Segmentation

Jianwei Gao[0000−0002−6358−4117], Juan Xu, and Honggao Fei

Digital Health China Technologies Co., LTD, Beijing, China
{gaojw,xujuan,feihg}@dchealth.com

**Abstract.** In this paper, we adopt a "pseudo-labeling" approach to semi-supervised learning based on 50 labeled images and 2000 unlabeled images. This approach yields a model with 0.7496 mean DSC on the validation set, outperforming the 0.6903 mean DSC of the model with only 50 labeled images.

**Keywords:** Abdominal organ segmentation · Semi-supervised · Pseudo-labeling.

## 1 Introduction

Abdomen organ segmentation has many important clinical applications. Typically, a large number of labeled data is required to train a accurate segmentation model. However, manually annotating organs from CT scans is time-consuming and labor-intensive. This requires us to use appropriate semi-supervised segmentation methods to use unlabeled data, such as disturbance regularization based on data or model [7] [8] and consistency constraint based on multitask [6].

FLARE22 provides 50 labeled images and 2000 unlabeled images to train the segmentation model of 13 organs. There are three main difficulties. First, we need to realize the segmentation of 13 organs. Second, more than 97% of the training data are unlabeled. Third, we need to balance model performance and resource consumption.

In order to use unlabeled data as well as labeled data, we adopted a pseudo-labeling approach to develop a segmentation model drawing on the idea of developing a classification model in [5]. Specifically, we first trained a model with labeled data, and then used the model to predict the unlabeled data to give them pseudo-labels. Finally, we fine-tuned the original model using all labeled data and partially filtered pseudo-labeled data.

## 2 Method

### 2.1 Preprocessing

We use several pre-processing strategies as follows.

- Cropping strategy
  We use the CT scans as the data source to generate the bounding box of foreground, and then crop only the foreground object of the images.
- Resampling method for anisotropic data
  We resample the original data to unify the voxel spacing into $[1.0, 1.0, 1.0]$.
- Intensity normalization method
  We normalize the intensity of $[-300.0, 300.0]$ to $[0.0, 1.0]$ and change those less than $-300.0$ and those greater than $300.0$ to $0.0$ and $1.0$, respectively.

### 2.2    Proposed Method

Figure 1 illustrates the applied 3D nnU-Net [4], where a 3D U-Net architecture is adopted. We use the leaky ReLU function with a negative slope of 0.01 as the activation function. Our 3D nnU-Net has 14 out channels, corresponding to the background and 13 organs respectively.

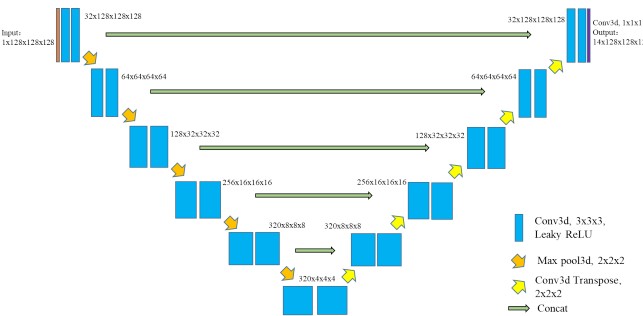

**Fig. 1.** Our 3D U-Net architecture

Our pseudo-labeling strategy for using unlabeled images is shown in Figure 2. First, we trained a model with the 50 labeled images. Then we used this model to predict the 2000 unlabeled images to give them pseudo-labels. After that, we picked out 676 pseudo-labeled images with at least 2000 voxels for each organ to ensure that each organ of each pseudo-labeled image is present and not too small, and put them together with the 50 labeled images. At last, we used these 726 images to fine-tune the original model.

We use the sum of Dice loss (after applying a softmax function) and Cross Entropy Loss as the loss function.

When predicting a single image with the trained segmentation model, we first resample it to a voxel spacing of [1.0, 1.0, 1.0], as we did during training, and try to predict. If there is a "CUDA out of memory" error, we resample it to [2.0, 2.0, 2.0] voxel spacing to reduce the size of the resampled image and thus reduce the usage of GPU memory.

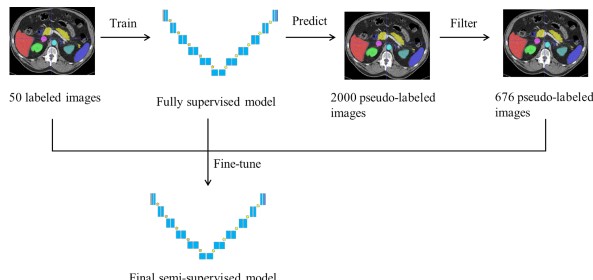

**Fig. 2.** Pseudo-labeling strategy

### 2.3 Post-processing

During model prediction, we select the label (from 0 to 13) corresponding to the largest of the 14 outputs for each voxel.

## 3 Experiments

### 3.1 Dataset and evaluation measures

The FLARE 2022 is an extension of the FLARE 2021 [9] with more segmentation targets and more diverse abdomen CT scans. The dataset is curated from more than 20 medical groups under the license permission, including MSD [11], KiTS [2,3], AbdomenCT-1K [10], and TCIA [1]. The training set includes 50 labelled CT scans with pancreas disease and 2000 unlabelled CT scans with liver, kidney, spleen, or pancreas diseases. The validation set includes 50 CT scans with liver, kidney, spleen, or pancreas diseases. The testing set includes 200 CT scans where 100 cases has liver, kidney, spleen, or pancreas diseases and the other 100 cases has uterine corpus endometrial, urothelial bladder, stomach, sarcomas, or ovarian diseases. All the CT scans only have image information and the center information is not available.

The evaluation measures consist of two accuracy measures: Dice Similarity Coefficient (DSC) and Normalized Surface Dice (NSD), and three running efficiency measures: running time, area under GPU memory-time curve, and area under CPU utilization-time curve. All measures will be used to compute the ranking. Moreover, the GPU memory consumption has a 2 GB tolerance.

### 3.2 Implementation details

**Environment settings** The development environments and requirements are presented in Table 1.

**Table 1.** Development environments and requirements.

| | |
|---|---|
| Windows/Ubuntu version | Ubuntu 20.04.4 LTS |
| CPU | Intel(R) Xeon(R) Gold 5218R CPU @ 2.10GHz |
| RAM | 128G |
| GPU (number and type) | 2 NVIDIA Tesla T4 (16G) |
| CUDA version | 11.6 |
| Programming language | Python 3.6 |
| Deep learning framework | Pytorch (Torch 1.10.1, torchvision 0.11.2) |
| Specific dependencies | numpy 1.19.5, SimpleITK 2.0.2, monai 0.8.1 |

**Training protocols** As described below.

Random flipping strategy (only for initial training stage): each image has a 20% probability of flipping along the x-axis and a 20% probability of flipping along the y-axis.

Random Gaussian smooth (only for initial training stage): each image has a 10% probability of being Gaussian smoothed with sigma in (0.5, 1.15) for every spatial dimension.

Random Gaussian noise (only for initial training stage): each image has a 10% probability of being added with Gaussian noise with mean in (0, 0.5) and standard deviation in (0, 1).

Random intensity change (only for initial training stage): each image has a 10% probability of changing intensity with gamma in (0.5, 2.5).

Random intensity shift (only for initial training stage): each image has a 10% probability of shifting intensity with offsets in (0, 0.3).

Patch sampling strategy: 2 patches of size $[128, 128, 128]$ are randomly cropped from each image. The center of each patch has 50% probability in the foreground and 50% probability in the background.

Optimal model selection criteria: we tried several different training protocols and selected the model with the highest DSC on the validation set.

Some details of the initial training stage and the fine-tuning stage are shown in Table 2 and Table 3 respectively.

## 4    Results and discussion

### 4.1    Quantitative results on validation set

DSC results on validation set are shown in Table 4. It can be seen from the table that the generalization ability of the model is indeed improved by using unlabeled data through the "pseudo-labeling" method.

### 4.2    Qualitative results on validation set

Two examples of good segmentation are shown in Figure 3 and two examples of bad segmentation are shown in Figure 4. Visualization is achieved with ITK-SNAP [12] version 3.8.0.

**Table 2.** Training protocols (initial training stage).

| | |
|---|---|
| Network initialization | "he" normal initialization |
| Batch size | 2 |
| Patch size | 128×128×128 |
| Total epochs | 1600 |
| Optimizer | Adam |
| Initial learning rate (lr) | 0.0001 |
| Lr decay schedule | initial learning rate$\times(1 - epoch/500)^{0.9}$ |
| Training time | 77 hours |
| Loss function | the sum of dice loss and cross entropy loss |
| Number of model parameters | 31.42M |

**Table 3.** Training protocols (fine-tuning stage).

| | |
|---|---|
| Network initialization | model after initial training |
| Batch size | 2 |
| Patch size | 128×128×128 |
| Total epochs | 40 |
| Optimizer | Adam |
| Initial learning rate (lr) | 0.00005 |
| Lr decay schedule | initial learning rate$\times(1 - epoch/500)^{0.9}$ |
| Training time | 39 hours |
| Loss function | the sum of dice loss and cross entropy loss |
| Number of model parameters | 31.42M |

**Table 4.** Results on validation set.

| | Without using unlabeled data | Using unlabeled data |
|---|---|---|
| Mean DSC | 0.6903 | 0.7496 |
| Liver | 0.9312 | 0.9493 |
| RK | 0.7151 | 0.8098 |
| Spleen | 0.8180 | 0.8962 |
| Pancreas | 0.6631 | 0.7506 |
| Aorta | 0.7474 | 0.7953 |
| IVC | 0.7003 | 0.7692 |
| RAG | 0.6792 | 0.6910 |
| LAG | 0.5257 | 0.5400 |
| Gallbladder | 0.6235 | 0.6543 |
| Esophagus | 0.6196 | 0.6641 |
| Stomach | 0.7550 | 0.8219 |
| Duodenum | 0.5261 | 0.5803 |
| LK | 0.6703 | 0.8225 |

From the perspective of organs, the segmentation results of organs with fewer surrounding organs are better, such as liver and spleen. From the perspective of images, some potential reasons for the bad-segmentation cases are listed below.

(1) The size of the case is very large, so we have to reduce the size of the case by resampling to avoid GPU memory overflow.

(2) The case is not clear, distorted, or skewed.

(3) There are rare structures in the case that are not in the training set.

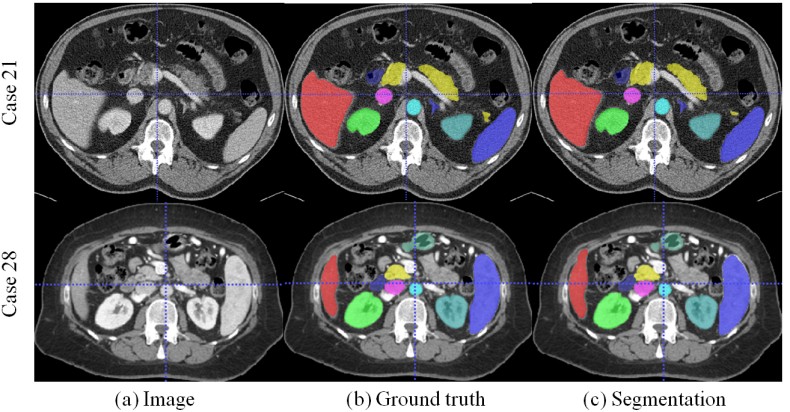

**Fig. 3.** Good segmentation examples

### 4.3   Segmentation efficiency results on validation set

Segmentation efficiency results for the 5th validation submission are shown in Table 5.

**Table 5.** Results on validation set.

| | |
|---|---|
| Running time | 1538.14 seconds |
| Maximal GPU Memory | 16327MB |
| Area under GPU memory-time curve | 11050890 |
| Area under CPU utilization-time curve | 26722.79 |

The running time is relatively short since we didn't use any cascaded framework. In the testing phase, we used the "sliding_window_inference" function of monai to slice the image into several 128×128×128 patches and predict them separately. This can lead to a large GPU memory consumption when the image size is large.

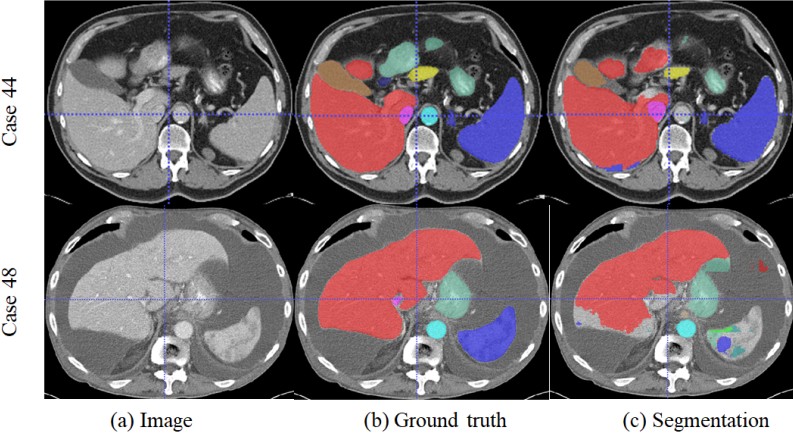

(a) Image         (b) Ground truth         (c) Segmentation

**Fig. 4.** Bad segmentation examples

## 4.4 Results on final testing set

DSC and NSD results on final testing set are shown in Table 6.

**Table 6.** Results on final testing set.

|  | DSC results | NSD results |
| --- | --- | --- |
| Mean | 0.7502 | 0.7779 |
| Liver | 0.9402 | 0.9005 |
| RK | 0.8230 | 0.7567 |
| Spleen | 0.8614 | 0.8052 |
| Pancreas | 0.7151 | 0.8071 |
| Aorta | 0.7971 | 0.8007 |
| IVC | 0.7663 | 0.7480 |
| RAG | 0.7484 | 0.8588 |
| LAG | 0.6396 | 0.7515 |
| Gallbladder | 0.6575 | 0.6231 |
| Esophagus | 0.6249 | 0.7233 |
| Stomach | 0.7860 | 0.7977 |
| Duodenum | 0.5739 | 0.7628 |
| LK | 0.8195 | 0.7769 |

### 4.5   Limitation and future work

In terms of model accuracy, first, we give pseudo-labels only once for the un-labeled images at present. In the future, we are going to give pseudo-labels and fine-tune the model for several times. Second, we used the same rules for all organs when filtering the pseudo-labeled images. It is more reasonable to use different rules for different organs. Third, we consider using some post-processing methods, such as largest connected component extraction, hole filling, open operation and closed operation, which are not used at present.

In terms of segmentation efficiency, we consider changing the value of the "device" parameter of the "sliding_window_inference" function of monai from "torch.device('cuda')" to "torch.device('cpu')" to reduce the GPU memory consumption. In addition, we consider using some optimization methods to improve the running speed of the model in the future.

## 5   Conclusion

Using unlabeled data through "pseudo-labeling" method can improve the performance of the model.

**Acknowledgements** The authors of this paper declare that the segmentation method they implemented for participation in the FLARE 2022 challenge has not used any pre-trained models nor additional datasets other than those provided by the organizers. The proposed solution is fully automatic without any manual intervention.

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
