# OpenReview forum: "A Pseudo-labeling Approach to Semi-supervised Organ Segmentation"
_MICCAI.org/2022/Challenge/FLARE_

### Official Review · Reviewer_ufzd · 2022-09-16
**Good try and better if the baseline of labeled data could be improved**

**Rating:** 6
**Confidence:** 3

**Review:**


Pros:
1. A base segmentation method for abdominal organs.
2. Bigdata and Pseudo-labeling learning is indeed useful for less annotation CT scans.

Cons:
1. The pixel values larger than 300.0 should be set into 300.0, and smaller than -300.0 should be -300.0.
2.  Advise to try coarse-to-fine framework.
3. Mixed precision work may be useful for your work

---

> ### Author Response · Authors · 2022-10-14
> **Dear reviewers, thank you very much for your comments.**
>
> 1. We did do that but the description in our manuscript is not accurate. In revised manuscript, we have corrected this error.
> 2. Thank you very much for your suggestions. We will try it in the futrue.
> 3. Thank you very much for your suggestions. We will try it in the future.
>
> We are sorry that it was too late to optimize our model before we submitted it on 17 July 2022. After that, we tried the methods mentioned in "Future Work". By using these methods, the mean DSC of 13 organs (of the 20 validation cases that the ground truth are given) reached 0.852.

---

### Official Review · Reviewer_FZE8 · 2022-09-19
**Average paper**

**Rating:** 5
**Confidence:** 4

**Review:**

Pros:
- Interesting augmentation strategy
- Good papers, clear tables

Cons
- No image describing the method
- Poor section layout in method description
- Not a very original idea

Most of all, poor DSC scores, both for baseline and for proposed method. The actual baseline scores, which other papers reached, are higher than the baseline scores presented here.

---

> ### Author Response · Authors · 2022-10-14
> **Dear reviewers, thank you very much for your comments.**
>
> 1.  In Method section, we have added Figure 2 to show our pseudo-labeling strategy.
> 2. We have adjusted the section layout in method description.
> 3. Our method is indeed not a very original idea.
> 4. We are sorry that it was too late to optimize our model before we submitted it on 17 July 2022. After that, we tried the methods mentioned in "Future Work". By using these methods, the mean DSC of 13 organs (of the 20 validation cases that the ground truth are given) reached 0.852.

---

### Official Review · Reviewer_AkNS · 2022-09-20
**Method is not novel, but results are not bad**

**Rating:** 6
**Confidence:** 5

**Review:**

Pros:
- good dice
- clear, simple method
- I would suggest to use this model as a baseline

Cons:
- method is not novel

---

> ### Author Response · Authors · 2022-10-14
> **Dear reviewers, thank you very much for your comments.**
>
> Our pseudo-labeling strategy is indeed not novel. After we submitted our model on 17 July 2022, we tried some methods mentioned in "Future Work", such as giving pseudo-labels and fine-tuning the model for several times, different rules for different organs when filtering the pseudo-labeled images.

---

### Official Review · Reviewer_NeA2 · 2022-09-20
**The method is clear but more detailed intuition could be helpful**

**Rating:** 6
**Confidence:** 4

**Review:**

Advantages:
1. The method description is clear.

Disadvantages:
1. A small description of the intuition behind the method could be helpful.
2. Neural networks are usally sensitive to the spacing. So, changing the spacing on the inference doesn't seem to be good idea. You could change spacing on training instead, change patch size or try to reimplement the patch-wise inference function in more efficient way.

---

> ### Author Response · Authors · 2022-10-14
> **Dear reviewers, thank you very much for your comments.**
>
> 1. In Introduction section, we have added more descriptions. In Method section, we have optimized the descriptions and added Figure 2 to
> show our pseudo-labeling strategy.
> 2. We also realized that this practice would reduce the prediction accuracy of the model. However, before we submitted our model on 17 July 2022, we hadn't found a better way to reduce the GPU memory consumption. After that, we found that we could reduce the maximal GPU memory from about 16G to about 3G using the method mentioned in "Future Work". Thus, we don't need to change the spacing at now.

---

### Meta-Review · Program_Chairs · 2022-09-28

**Recommendation:** Major Revision
**Confidence:** 5

**Metareview:**

Figure 1 Suspected plagiarism
Reviewers raise many concerns and suggestions. Please address all comments in the revised manuscript.

---

> ### Author Response · Authors · 2022-10-14
> **Dear FLARE Program Chairs, thank you very much for your comments**
>
> In revised manuscript, we have redrawn Figure 1.